# When does return-conditioned supervised learning work for offline reinforcement learning?

**David Brandfonbrener**
New York University
david.brandfonbrener@nyu.edu

**Alberto Bietti**
New York University

**Jacob Buckman**
MILA

**Romain Laroche**
Microsoft Research

**Joan Bruna**
New York University

## Abstract

Several recent works have proposed a class of algorithms for the offline reinforcement learning (RL) problem that we will refer to as return-conditioned supervised learning (RCSL). RCSL algorithms learn the distribution of actions conditioned on both the state and the return of the trajectory. Then they define a policy by conditioning on achieving high return. In this paper, we provide a rigorous study of the capabilities and limitations of RCSL, something which is crucially missing in previous work. We find that RCSL returns the optimal policy under a set of assumptions that are stronger than those needed for the more traditional dynamic programming-based algorithms. We provide specific examples of MDPs and datasets that illustrate the necessity of these assumptions and the limits of RCSL. Finally, we present empirical evidence that these limitations will also cause issues in practice by providing illustrative experiments in simple point-mass environments and on datasets from the D4RL benchmark.

## 1 Introduction

In recent years, deep learning has proven to be an exceptionally powerful generic algorithm for solving supervised learning (SL) tasks. These approaches tend to be stable, and scale well with compute and data [17]. In contrast, deep reinforcement learning algorithms seem to lack these nice properties; results are well known to be sensitive to hyperparameters and difficult to replicate. In spite of this, deep reinforcement learning (RL) has achieved impressive feats, such as defeating human champions at Go [25]. This juxtaposition of success and instability has inspired researchers to explore alternative approaches to reinforcement learning that more closely resemble supervised learning in hopes of making deep RL as well-behaved as deep SL.

One family of algorithms that has garnered great interest recently is return-conditioned supervised learning (RCSL). The core idea of RCSL is to learn the return-conditional distribution of actions in each state, and then define a policy by sampling from the distribution of actions that receive high return. This was first proposed for the online RL setting by work on Upside Down RL [23, 26] and Reward Conditioned Policies [21]. The idea was extended to the offline RL setting using transformers that condition on the entire history of states rather than just the current Markovian state in the Decision Transformer (DT) work [8, 12]. Recent work on RL via Supervised Learning (RvS) [9] unifies and simplifies ideas from these prior works with ideas about goal-conditioned policies.

Importantly, none of this prior work provides theoretical guarantees or analysis of the failure modes of the return-conditioning approach. In contrast, the more established dynamic programming (DP) algorithms for RL are better understood theoretically. This paper attempts to address this gap in

36th Conference on Neural Information Processing Systems (NeurIPS 2022).

understanding, in order to assess when RCSL is a reliable approach for offline RL. Specifically, we answer the following questions:

- What optimality guarantees can we make for RCSL? Under what conditions are they necessary and sufficient?
- In what situations does RCSL fail in theory and in practice?
- How does RCSL relate to other approaches, such as DP and behavior cloning (BC)?

We find that although RCSL does select a near-optimal policy under certain conditions, the necessary assumptions are more strict than those for DP. In particular, RCSL (but not DP) requires nearly deterministic dynamics in the MDP, knowledge of the proper value to condition on, and for the conditioning value to be supported by the distribution of returns in the dataset. We provide simple tabular examples to demonstrate the necessity of these assumptions. The shortcomings of RCSL that we identify in theory are verified empirically with some simple experiments using neural models on ad-hoc example problems as well as benchmark datasets. We conclude that RCSL alone is unlikely to be a general solution for offline RL problems, but does show promise in some specific situations such as deterministic MDPs with high-quality behavior data.

## 2 Preliminaries

### 2.1 Setup

We will consider an offline RL setup where we are given a dataset $\mathcal{D}$ of trajectories $\tau = (o_1, a_1, r_1, \cdots, o_H, a_H, r_H)$ of observations $o_t \in \mathcal{O}$, actions $a_t \in \mathcal{A}$, and rewards $r_t \in [0, 1]$ generated by some behavior policy $\beta$ interacting with a finite horizon MDP with horizon $H$. Let $g(\tau) = \sum_{t=1}^{H} r_t$ denote the cumulative return of the trajectory (we will just use $g$ when the trajectory is clear from context). And let $J(\pi) = \mathbb{E}_{\tau \sim \pi}[g(\tau)]$ be the expected return of a policy $\pi$. We then let the state representation $s_t \in \mathcal{S}$ be any function of the history of observations, actions, and rewards up to step $t$ along with $o_t$. To simplify notation in the finite horizon setting, we will sometimes drop the timestep from $s$ to refer to generic states and assume that we can access the timestep from the state representation as $t(s)$. Let $P_\pi$ denote the joint distribution over states, actions, rewards, and returns induced by any policy $\pi$.

In this paper, we focus on the RCSL approach that learns by return-conditioned supervised learning. Explicitly, at training time this method minimizes the empirical negative log likelihood loss:

$$\hat{L}(\pi) = -\sum_{\tau \in \mathcal{D}} \sum_{1 \leq t \leq H} \log \pi(a_t | s_t, g(\tau)). \tag{1}$$

Then at test time, an algorithm takes the learned policy $\pi$ along with a conditioning function $f(s)$ to define the test-time policy $\pi_f$ as:

$$\pi_f(a|s) := \pi(a|s, f(s)). \tag{2}$$

Nota bene: the Decision Transformer [8] is captured in this framework by defining the state space so that the state $s_t$ at time $t$ also contains all past $o_{t'}$, $a_{t'}$, and $r_{t'}$ for $t' < t$. In prior work, $f$ is usually chosen to be a constant at the initial state and to decrease with observed reward along a trajectory, which is captured by a state representation that includes the history of rewards.

### 2.2 The RCSL policy

To better understand the objective, it is useful to first consider its optimum in the case of infinite data. It is clear that our loss function attempts to learn $P_\beta(a|s, g)$ where $\beta$ is the behavior policy that generated the data (and recall that $P_\beta$ refers to the distribution over states, actions, and returns induced by $\beta$). Factoring this distribution, we quickly see that the optimal policy $\pi_f^{\text{RCSL}}$ for a specific conditioning function $f$ can be written as:

$$\pi_f^{\text{RCSL}}(a|s) = P_\beta(a|s, f(s)) = \frac{P_\beta(a|s)P_\beta(f(s)|s, a)}{P_\beta(f(s)|s)} = \beta(a|s)\frac{P_\beta(f(s)|s, a)}{P_\beta(f(s)|s)}. \tag{3}$$

Essentially, the RCSL policy re-weights the behavior based on the distribution of future returns.

**Connection to distributional RL.** In distributional RL [4], the distribution of future returns under a policy $\pi$ from state $s$ and action $a$ is defined as: $G^\pi(s,a) \sim g = \sum_{t=t(s)}^{H} r_t \mid \tau \sim \pi, s_{t(s)} = s, a_{t(s)} = a$. The RCSL policy is precisely proportional to the product of the behavior policy and the density of the distributional Q function of the behavior policy (i.e. $P_\beta(g|s,a)$).

## 2.3 Related work

As noted in the introduction, our work is in direct response to the recent line of literature on RCSL [23, 26, 21, 8, 12, 9]. Specifically, we will focus on the DT [8] and RvS [9] formulations in our experiments since they also focus on the offline RL setting. Note that another recent work introduced the Trajectory Transformer [15] which does not fall under the RCSL umbrella since it performs planning in the learned model to define a policy.

Another relevant predecessor of RCSL comes from work on goal-based RL [16]. Compared to RCSL, this line of work replaces the target return $g$ in the empirical loss function by a goal state. One instantiation is hindsight experience replay (HER) where each trajectory in the replay buffer is relabeled as if the observed final state was in fact the goal state [2]. Another instance is goal-conditioned supervised learning [GCSL, 13], which provides more careful analysis and guarantees, but the guarantees (1) are not transferable to the return-conditioned setting, (2) assume bounds on $L_\infty$ errors in TV distance instead of dealing with expected loss functions that can be estimated from data, and (3) do not provide analysis of the tightness of the bounds.

Concurrent work [34, 22, 33] also all raise the issue of RCSL in stochastic environments with infinite data, and present some algorithmic solutions. However, none of this work addresses the potentially more fundamental issue of sample complexity that arises from the requirement of return coverage that we discuss in Section 4.

## 3 When does RCSL find the optimal policy?

We begin by exploring how RCSL behaves with infinite data and a fully expressive policy class. In this setting, classic DP algorithms (e.g. Q-learning) are guaranteed to converge to the optimal policy under coverage assumptions [27]. But we now show that this is not the case for RCSL, which requires additional assumptions for a similar guarantee. Our approach is to first derive a positive result: under certain assumptions, the policy which optimizes the RCSL objective (Section 2.2) is guaranteed to be near-optimal. We then illustrate the limitations of RCSL by providing simple examples that are nonetheless challenging for these methods in order to demonstrate why our assumptions are necessary and that our bound is tight.

**Theorem 1** (Alignment with respect to the conditioning function). *Consider an MDP, behavior $\beta$ and conditioning function $f$. Assume the following:*

1. *Return coverage: $P_\beta(g = f(s_1)|s_1) \geq \alpha_f$ for all initial states $s_1$.*

2. *Near determinism: $P(r \neq r(s,a)$ or $s' \neq T(s,a)|s,a) \leq \epsilon$ at all $s, a$ for some functions $T$ and $r$. Note that this does not constrain the stochasticity of the initial state.*

3. *Consistency of $f$: $f(s) = f(s') + r$ for all $s$.[1]*

*Then*

$$\mathbb{E}_{s_1}[f(s_1)] - J(\pi_f^{RCSL}) \leq \epsilon \left( \frac{1}{\alpha_f} + 2 \right) H^2. \tag{4}$$

*Moreover, there exist problems where the bound is tight up to constant factors.*

The proof is in Appendix C.1. Note that the quantity $\mathbb{E}_{s_1}[f(s_1)]$ is specific to the structure of RCSL algorithms and captures the notion that the ideal RCSL policy will be able to reproduce policies of any value when given different conditioning functions (with appropriate data). The theorem immediately yields the following corollaries (with proof in Appendix C.1).

---

[1]Note this can be exactly enforced (as in prior work) by augmenting the state space to include the cumulative reward observed so far.

**Corollary 1.** *Under the assumptions of Theorem 1, there exists a conditioning function $f$ such that*

$$J(\pi^*) - J(\pi_f^{RCSL}) \leq \epsilon \left( \frac{1}{\alpha_f} + 3 \right) H^2. \tag{5}$$

**Corollary 2.** *If $\alpha_f > 0$, $\epsilon = 0$, and $f(s_1) = V^*(s_1)$ for all initial states $s_1$, then $J(\pi_f^{RCSL}) = J(\pi^*)$.*

The corollaries tell us that in near determinisitc environments with the proper conditioning functions and data coverage, it is possible for RCSL to recover near optimal policies. These assumptions are somewhat strong compared to those needed for DP-based approaches, so we will now explain why they are necessary for our analysis.

**Tightness.** To demonstrate tightness we will consider the simple examples in Figure 1. These MDPs and behavior policies demonstrate tightness in $\epsilon$ and $\alpha_f$ up to constant factors, and provide insight into how stochastic dynamics lead to suboptimal behavior from RCSL algorithms.

(a) An example where the bound is tight. $\mathcal{B}$ denotes the Bernoulli distribution.

(b) An example where RCSL also has large regret.

(c) An example where RCSL also has large regret for any conditioning function.

Figure 1: Failure modes of RCSL in stochastic environments with infinite data.

First, consider the example in Figure 1a with conditioning $f(s_1) = 1$. There is only one possible policy in this case, and it has $J(\pi) = \epsilon$ so that $\mathbb{E}[f(s_1)] - J(\pi) = 1 - \epsilon$. Note that $\alpha_f = \epsilon$, so we have that $\epsilon/\alpha_f = 1$. Thus, the bound is tight in $\epsilon/\alpha_f$. This example shows that the goal of achieving a specific desired return is incompatible with stochastic environments.

This first example is somewhat silly since there is only one action, so the learned policy does not actually suffer any regret. To show that this issue can in fact lead to regret, consider the example in Figure 1b, again with conditioning $f(s_1) = 1$. Then applying the reasoning from Section 2.2,

$$\pi_f^{RCSL}(a_1|s_1) = \beta(a_1|s_1) \frac{P_\beta(g=1|s_1,a_1)}{P_\beta(g=1|s_1)} = 0.5 \cdot \frac{0}{0.5 \cdot \epsilon} = 0. \tag{6}$$

So we get that $\mathbb{E}[f(s_1)] - J(\pi_f^{RCSL}) = 1 - \epsilon$, while $\epsilon/\alpha_f = \epsilon/(\epsilon/2) = 2$ (which is on the same order, up to a constant factor). However, in this case the learned policy $\pi_f^{RCSL}$ suffers substantial regret since the chosen action $a_2$ has substantially lower expected value than $a_1$ by $1 - 2\epsilon$.

The issue in the second example could be resolved by changing the conditioning function so that $f(s_1) = 1 - \epsilon$. Now we will consider the example in Figure 1c where we will see that there exist cases where the bias of RCSL in stochastic environments can remain regardless of the conditioning function. In this MDP, the only returns that are supported are $g = 0$ or $g = 1$. For $f(s_1) = 1$, plugging in to the formula for $\pi_f$ yields

$$\pi_f^{RCSL}(a_1|s_1) = \beta(a_1|s_1) \frac{P_\beta(g=1|s_1,a_1)}{P_\beta(g=1|s_1)} = \epsilon \frac{1-\epsilon}{\epsilon(1-\epsilon)+(1-\epsilon)\epsilon} = \frac{1}{2}. \tag{7}$$

Thus, $\mathbb{E}[f(s_1)] - J(\pi_f^{RCSL}) = 1/2$ and $J(\pi^*) - J(\pi_f^{RCSL}) = 1/2 - \epsilon$. This shows that merely changing the conditioning function is not enough to overcome the bias of the RCSL method in stochastic environments.

These examples show that even for MDPs that are $\epsilon$-close to being deterministic, the regret of RCSL can be large. But, in the special case of deterministic MDPs we find that RCSL can indeed recover the optimal policy. And note that we still allow for stochasticity in the initial states in these deterministic MDPs, which provides a rich setting for problems like robotics that requires generalization over the state space from finite data. In the next section, we will consider more precisely what happens to RCSL algorithms with finite data and limited model classes.

**Trajectory stitching.** Another issue often discussed in the offline RL literature is the idea of trajectory stitching [31, 8]. Ideally, an offline RL agent can take suboptimal trajectories that overlap and stitch them into a better policy. Clearly, DP-based algorithms can do this, but it is not so clear that RCSL algorithms can. In Appendix B we provide theoretical and empirical evidence that in fact they cannot perform stitching in general, even with infinite data. While this does not directly affect our bounds, the failure to perform stitching is an issue of practical importance for RCSL methods.

## 4 Sample complexity of RCSL

Now that we have a better sense of what policy RCSL will converge to with infinite data, we can consider how quickly (and under what conditions) it will converge to the policy $\pi_f$ when given finite data and a limited policy class, as will occur in practice. We will do this via a reduction from the regret relative to the infinite data solution $\pi_f$ to the expected loss function $L$ minimized at training time by RCSL, which is encoded in the following theorem.

**Theorem 2** (Reduction of RCSL to SL). *Consider any function $f : \mathcal{S} \to \mathbb{R}$ such that the following two assumptions hold:*

1. *Bounded occupancy mismatch:* $\frac{P_{\pi_f^{RCSL}}(s)}{P_\beta(s)} \leq C_f$ *for all $s$.*

2. *Return coverage:* $P_\beta(g = f(s)|s) \geq \alpha_f$ *for all $s$.*

*Define the expected loss as $L(\hat{\pi}) = \mathbb{E}_{s \sim P_\beta} \mathbb{E}_{g \sim P_\beta(\cdot|s)}[D_{\mathrm{KL}}(P_\beta(\cdot|s,g)\|\hat{\pi}(\cdot|s,g))]$. Then for any estimated RCSL policy $\hat{\pi}$ that conditions on $f$ at test time (denoted by $\hat{\pi}_f$), we have that*

$$J(\pi_f^{RCSL}) - J(\hat{\pi}_f) \leq \frac{C_f}{\alpha_f} H^2 \sqrt{2L(\hat{\pi})}. \tag{8}$$

The proof can be found in Appendix C.3. Note that we require a similar assumption of return coverage as before to ensure we have sufficient data to define $\pi_f$. We also require an assumption on the state occupancy of the idealized policy $\pi_f$ relative to $\beta$. This assumption is needed since the loss $L(\hat{\pi})$ is optimized on states sampled from $P_\beta$, but we care about the expected return of the learned policy relative to that of $\pi_f$, which can be written as an expectation over states sampled from $P_{\pi_f}$.

This gives us a reduction to supervised learning, but to take this all the way to a sample complexity bound we need to control the loss $L(\hat{\pi})$ from finite samples. Letting $N$ denote the size of the dataset, the following corollary uses standard uniform convergence results from supervised learning [24] to yield finite sample bounds.

**Corollary 3** (Sample complexity of RCSL). *To get finite data guarantees, add to the above assumptions the assumptions that (1) the policy class $\Pi$ is finite, (2) $|\log \pi(a|s,g) - \log \pi(a'|s',g')| \leq c$ for any $(a,s,g,a',s',g')$ and all $\pi \in \Pi$, and (3) the approximation error of $\Pi$ is bounded by $\epsilon_{approx}$, i.e. $\min_{\pi \in \Pi} L(\pi) \leq \epsilon_{approx}$. Then with probability at least $1 - \delta$,*

$$J(\pi_f^{RCSL}) - J(\hat{\pi}_f) \leq O\left(\frac{C_f}{\alpha_f} H^2 \left(\sqrt{c}\left(\frac{\log |\Pi|/\delta}{N}\right)^{1/4} + \sqrt{\epsilon_{approx}}\right)\right). \tag{9}$$

The proof is in Appendix C.4. Analyzing the bound, we can see that the dependence on $N$ is in terms of a fourth root rather than the square root, but this comes from the fact that we are optimizing a surrogate loss. Namely the learner optimizes KL divergence, but we ultimately care about regret which we access by using the KL term to bound a TV divergence and thus lose a square root factor. A similar rate appears, for example, when bounding 0-1 classification loss of logistic regression [3, 5].

This corollary also tells us something about how the learner will learn to generalize across different values of the return. If the policy class is small (for some notion of model complexity) and sufficiently structured, then it can use information from the given data to generalize across values of $g$, using low-return trajectories to inform the model on high-return trajectories.

Note that a full sample complexity bound that competes with the optimal policy can be derived by combining this result with Corollary 1 as follows:

**Corollary 4** (Sample complexity against the optimal policy). *Under all of the assumptions of Corollary 1 and Corollary 3 we get:*

$$J(\pi^*) - J(\hat{\pi}_f) \leq O\left(\frac{C_f}{\alpha_f}H^2\left(\sqrt{c}\left(\frac{\log|\Pi|/\delta}{N}\right)^{1/4} + \sqrt{\epsilon_{approx}}\right) + \frac{\epsilon}{\alpha_f}H^2\right). \qquad (10)$$

**Tightness.** To better understand why the dependence on $1/\alpha_f$ is tight and potentially exponential in the horizon $H$, even in deterministic environments, we offer the example in Figure 2. Specifically, we claim that any value of $f(s_1)$ where the policy $\pi_f^{\text{RCSL}}$ prefers the good action $a_1$ from $s_1$ will require on the order of $10^{H/2}$ samples in expectation to recover as $\hat{\pi}_f$[2].

To see why this is the case, we consider the MDP illustrated in Figure 2 with horizon $H \gg 4$. The MDP has four states each with two actions. All transitions and rewards are deterministic. The only actions with non-zero reward are $r(s_2, a_1) = 1$ and $r(s_3, a_1) = 0.5$. The interesting decision is at $s_1$ where $a_1$ is better than $a_2$.

Note that for any integer $1 \leq k < H/2$, we have that $P_\beta(g = k|s_1, a_2) = 0.5 \cdot 0.5^{2k} = 0.5 \cdot (0.25)^k$, while $P_\beta(g = k|s_1, a_1) = 0.5 \cdot (0.1)^k$. Conditioning on any such $k$ will make us more likely to choose the bad action $a_2$ from $s_1$. The only way to increase the likelihood of the good action $a_1$ from $s_1$ and $s_2$ is to condition on $f(s_1) > H/2$. Unfortunately for RCSL, the probability of observing $g > H/2$ is extremely small, since for any such $f$ we have $P_\beta(g = f(s_1)) \leq 0.5 \cdot (0.1)^{H/2} \leq 10^{-H/2}$. Thus, both $\alpha_f$ and the sample complexity of learning for any $f$ that will yield a policy better than the behavior is exponential in the horizon $H$.

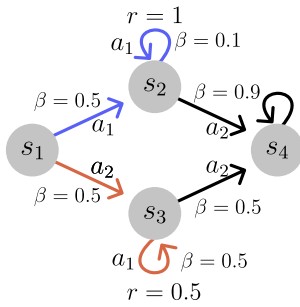

Figure 2: An example where RCSL has exponential sample complexity in a deterministic environment.

Fundamentally, the problem here is that RCSL uses trajectory-level information instead of performing dynamic programming on individual transitions. But, collecting enough trajectory-level information can take exponentially many samples in the horizon. In contrast, DP merely requires coverage of transitions in the MDP to perform planning and thus avoids this issue of exponential sample complexity. In the next section we will delve deeper into this comparison with DP-based approaches as well as the simple top-% BC baseline.

## 5 Comparing RCSL with bounds for alternative methods

Now that we understand the rate at which we expect RCSL to converge, we briefly present the convergence rates of two baseline methods for comparison. In particular, we will leverage an existing analysis of a DP-based algorithm, and conduct a novel analysis of top-% BC. We find that the sample complexity of RCSL has a similar rate to top-% BC, and is worse than DP due to the potentially exponential dependence on horizon that stems from return coverage.

### 5.1 Comparison to dynamic programming.

We will compare to the state of the art (to our knowledge) bound for a DP-based offline RL algorithm. Namely, we will look at the results of [32] for pessimistic soft policy iteration. Similar results exist for slightly different algorithms or assumptions [7, 30], but we choose this one since it is both the tightest and more closely aligns with the practical actor-critic algorithms that we use for our experiments. Their bound makes the following assumptions about the function class $F$ and the dataset (letting $\mathcal{T}^\pi$ represent the Bellman operator for policy $\pi$):

1. Realizability: for any policies $\pi, \pi'$ there exists $f \in F$ with $\|f - \mathcal{T}^\pi f\|_{2,P_{\pi'}}^2 \leq \epsilon_1$.

2. Bellman completeness: for any $\pi$ and $f \in F$ there exists $f' \in F$ such that $\|f' - \mathcal{T}^\pi f\|_{2,P_\beta}^2 \leq \epsilon_2$.

3. Coverage: $\frac{P_{\pi^*}(s,a)}{P_\beta(s,a)} \leq C$ for all $s, a$[3].

---

[2]Except for $f(s_1) = 0$, which will yield a policy substantially worse than the behavior.

[3]The original paper uses a slightly tighter notion of coverage, but this bound will suffice for our comparison.

With these assumptions in place, the sample complexity bound takes the form[4]:

$$J(\pi^*) - J(\hat{\pi}) \leq O\left(H^2\left(\sqrt{\frac{C\log|F||\Pi|/\delta}{N}}\right) + H^2\sqrt{C(\epsilon_1 + \epsilon_2)}\right) \qquad (11)$$

Note: this is the result for the "information-theoretic" form of the algorithm that cannot be efficiently implemented. The paper also provides a "practical" version of the algorithm for which the bound is the same except that the the square root in the first term is replaced with a fifth root.

There are several points of comparison with our analysis (specifically, our Corollary 4). The first thing to note is that for RCSL to compete with the optimal policy, we require nearly deterministic dynamics and a priori knowledge of the optimal conditioning function. These assumptions are not required for the DP-based algorithm; this is a critical difference, since it is clear that these conditions often do not hold in practice.

Comparing the coverage assumptions, our $C_f$ becomes nearly equivalent to $C$. The major difference is that our analysis of RCSL also requires dependence on return coverage $1/\alpha_f$. This is problematic since as seen in Section 4, this return coverage dependence can be exponential in horizon in cases where the state coverage does not depend on horizon.

Comparing the approximation error assumptions, we see that the realizability and completeness assumptions required for DP are substantially less intuitive than the standard supervised learning approximation error assumption needed for RCSL. These assumptions are not directly comparable, but intuitively the RCSL approximation error assumption is simpler.

Finally, dependence on $H$ is the same for both methods and dependence on $N$ depends on which version of the DP algorithm we compare to. For the information-theoretic algorithm DP has better dependence on $N$, but for the practical algorithm RCSL has better dependence. It is not clear whether the dependence on $N$ in either the RCSL analysis or in the analysis of the practical algorithm from [32] is tight, and it is an interesting direction for future work to resolve this issue.

## 5.2 Comparison to top-% behavior cloning.

The closest algorithm to RCSL is top-% BC, which was introduced as a baseline for Decision Transformers [8]. This algorithm simply sorts all trajectories in the dataset by return and takes the top $\rho$ fraction of trajectories to run behavior cloning (for $\rho \in [0, 1]$). The most obvious difference between this algorithm and RCSL is that RCSL allows us to plug in different conditioning functions at test time to produce different policies, while top-% BC learns only one policy. However, if we want to achieve high returns, the two algorithms are quite similar.

The full statements and proofs of our theoretical results for top-% BC are deferred to Appendix C.5. The results are essentially the same as those for RCSL except for two key modifications:

**Defining coverage.** The first difference in the analysis is the notion of coverage. For RCSL we needed the return distribution to cover the conditioning function $f$. For top-% BC we instead let $g_\rho$ be the $1 - \rho$ quantile of the return distribution over trajectories sampled by the behavior $\beta$ and then define coverage as $P_\beta(g \geq g_\rho|s) \geq \alpha_\rho$ for all $s$. This modification is somewhat minor.

**Sample size and generalization.** The main difference between RCSL and top-% BC is that the RCSL algorithm attempts to transfer information gained from low-return trajectories while the top-% BC algorithm simply throws those trajectories away. This shows up in the formal bounds since for a dataset of size $N$ the top-% BC algorithm only uses $\rho \cdot N$ samples while RCSL uses all $N$. Depending on the data distribution, competing with the optimal policy may require setting $\rho$ very close to zero (exponentially small in $H$) yielding poor sample complexity.

These bounds suggest that RCSL can use generalization across returns to provide improvements in sample complexity over top-% BC by leveraging all of the data. However, the RCSL model is attempting to learn a richer class of functions that conditions on reward, which may require a larger policy class negating some of this benefit. Overall, RCSL should expect to beat top-% BC if the behavior policy is still providing useful information about how to interact with the environment in low-return trajectories that top-% BC would throw away.

---

[4]The original paper considers an infinite horizon discounted setting. For the purposes of comparison, we will just assume that $\frac{1}{1-\gamma}$ can be replaced by $H$.

# 6 Experiments

We have illustrated through theory and some simple examples when we expect RCSL to work, but the theory does not cover all cases that are relevant for practice. In particular, it is not clear how the neural networks trained in practice can leverage generalization across returns. Moreover, one of the key benefits to RCSL approaches (as compared to DP) is that by avoiding the instabilities of non-linear off-policy DP in favor of supervised learning, one might hope that RCSL is more stable in practice. In this section we attempt to test these capabilities first through targeted experiments in a point-mass environment and then by comparisons on standard benchmark data.

Throughout this section we will consider six algorithms, two from each of three categories:

1. Behavior cloning (BC): standard behavior cloning (BC) and percentage behavior cloning (%BC) that runs BC on the trajectories with the highest returns [8].

2. Dynamic programming (DP): TD3+BC [11] a simple DP-based offline RL approach and IQL [20] a more stable DP-based offline RL approach.

3. Return-conditioned supervised learning (RCSL): RvS [9] an RCSL approach using simple MLP policies, and DT [8] an RCSL approach using transformer policies.

All algorithms are implemented in JAX [6] using flax [14] and the jaxrl framework [19], except for DT which is taken from the original paper. Full details can be found in Appendix D and code can be found at `https://github.com/davidbrandfonbrener/rcsl-paper`.

## 6.1 Point-mass datasets

First, we use targeted experiments to demonstrate how the tabular failure modes illustrated above can arise even in simple deterministic MDPs that may be encountered in continuous control. Specifically, we will focus on the issue of exponential sample complexity discussed in Section 4. We build our datasets in an environment using the Deepmind control suite [28] and MuJoCo simulator [29]. The environment consists of a point-mass navigating in a 2-d plane.

To build an example with exponential sample complexity we construct a navigation task with a goal region in the center of the environment. The dataset is constructed by running a behavior policy that is a random walk that is biased towards the top right of the environment. To construct different levels of reward coverage, we consider the environment and dataset under three different reward functions (ordered by probability of seeing a trajectory with high return, from lowest to highest):

(a) The "ring of fire" reward. This reward is 1 within the goal region, -1 in the ring of fire region surrounding the goal, and 0 otherwise

(b) The sparse reward. This reward is 1 within the goal region and 0 otherwise.

(c) The dense reward. This reward function is 1 within the goal region and gradually decays with the Euclidean distance outside of it.

Intuitively, the ring of fire reward will cause serious problems for RCSL approaches when combined with the random walk behavior policy. The issue is that any random walk which reached the goal region is highly likely to spend more time in the region of negative rewards than in the actual goal states, since the ring of fire has larger area than the goal. As a result, while they are technically supported by the distribution, it is unlikely to find many trajectories (if any at all) with positive returns in the dataset, let alone near-optimal returns. As a result, the RCSL-based approaches are not even able to learn to achieve positive returns, as seen in Figure 3.

The sparse reward is also difficult for the RCSL-based algorithms, for similar reasons; however the problem is less extreme since any trajectory that gets positive reward must go to the goal, so there is signal in the returns indicating where the goal is. In contrast, the dense reward provides enough signal in the returns that RCSL approaches are able to perform well, although still not as well as IQL. It is also worth noting that because the datset still does not have full coverage of the state-space, simple DP-based algorithms like TD3+BC can struggle with training instability.

## 6.2 Benchmark data

In addition to our targeted experiments we also ran our candidate algorithms on some datasets from the D4RL benchmark [10]. These are meant to provide more realistic and larger-scale data scenarios.

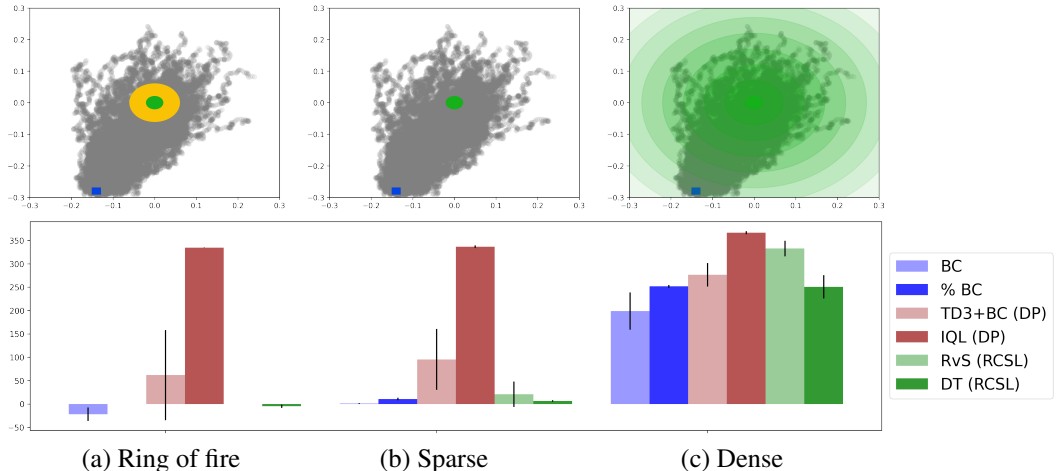

(a) Ring of fire        (b) Sparse        (c) Dense

Figure 3: RCSL fails under reward functions that lead to exponentially small probability of sampling good trajectories, but can generalize when the reward is dense. Error bars show standard deviation across three seeds. BC methods are in blue, DP methods in brown, and RCSL methods in green.

While this also makes these experiments less targeted, we can still see that the insights that we gained in simpler problems can be useful in these larger settings. We attempt to choose a subset of the datasets with very different properties from eachother. For example, the play data on the ant-maze environment is very diverse and plentiful while the human demonstration data on the pen environment has poor coverage but high values. Results are shown in Figure 4. And additional results leading to similar conclusions can be found in Appendix A.

We find that for most of the datasets DP-based algorithms TD3+BC and IQL outperform both the BC-based algorithms and RCSL-based algorithms. This is especially stark on the ANTMAZE datasets where the behavior policy is highly stochastic, requiring the learner to stitch together trajectories to achieve good performance. While none of these tasks has stochastic dynamics, the issues of return coverage and trajectory stitching persist.

In contrast, RCSL performs well when the behavior policy is already high quality, but not optimal (as in the PEN-HUMAN task). Since the data is suboptimal and reward is dense, there is opportunity for RCSL to outperform the BC-based methods. Moreover, since the data has poor coverage, standard DP approaches like TD3+BC are highly unstable.

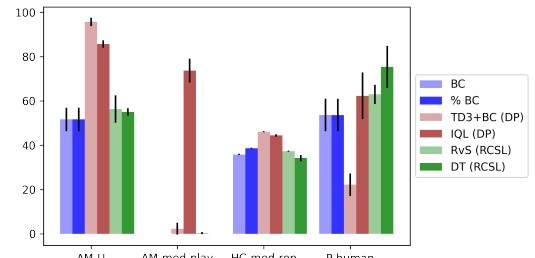

Figure 4: Data from ANTMAZE-UMAZE, ANTMAZE-MEDIUM-PLAY, HALFCHEETAH-MEDIUM-REPLAY, and PEN-HUMAN. Error bars show standard deviation across three seeds. Each algorithm is tuned over 4 values and best performance is reported.

IQL is more stable and performs similarly to the RCSL-based algorithms, but is outperformed by DT (perhaps due to the use of history-dependent policies).

## 7   Discussion

Looking back at our results, we can better place RCSL in relation to the more classical BC and DP algorithms. Like BC, RCSL relies on supervised learning and thus inherits its simplicity, elegance, and ease of implementation and debugging. However, it also inherits BC's dependence on the quality of the behavior policy. This dependence can be somewhat reduced in (nearly) deterministic environments, where conditioning on high returns can break the bias towards the behavior policy. But, the reliance on trajectory-level information still means that RCSL is fundamentally limited by

the quality of the best trajectories in the dataset, which can require a sample complexity exponential in horizon in order to compete with the optimal policy, even in deterministic environments.

In contrast, DP methods are capable of learning good policies even when the dataset does not contain any high-return trajectories and the environment is stochastic. This represents a fundamental gap between the two approaches that cannot be bridged within the RCSL paradigm. However, empirically, current deep DP algorithms are not well-behaved. These algorithms are often unstable and difficult to debug, although recent work has started to alleviate these issues somewhat [20].

In sum, for tasks where the requirements for RCSL to perform well are met, it is an excellent practical choice, with great advantages in simplicity over DP. Since many real-world tasks of relevance have these attributes, RCSL techniques could have substantial impact. But as a general learning paradigm, RCSL is fundamentally limited in ways that DP is not.

### Acknowledgments

This work was partially supported by NSF RI-1816753, NSF CAREER CIF 1845360, NSF CHS-1901091, NSF Scale MoDL DMS 2134216, Capital One and Samsung Electronics. DB was supported by the Department of Defense (DoD) through the National Defense Science & Engineering Graduate Fellowship (NDSEG) Program.

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
